# Sieve Search Centroiding Algorithm for Star Sensors

**DOI:** 10.3390/s23063222

**Published:** 2023-03-17

**Authors:** Vivek Chandran Karaparambil, Narayan Suresh Manjarekar, Pravin Madanrao Singru

**Affiliations:** Birla Institute of Technology and Science, Pilani, K. K. Birla Goa Campus, Sancoale 403726, Goa, India; narayan@goa.bits-pilani.ac.in (N.S.M.); pmsingru@goa.bits-pilani.ac.in (P.M.S.)

**Keywords:** aeronautical and space sensor systems, star sensor, sub-pixel, star centroiding, attitude estimation, sensor model analysis

## Abstract

The localization of the center of the star image formed on a sensor array directly affects attitude estimation accuracy. This paper proposes an intuitive self-evolving centroiding algorithm, termed the sieve search algorithm (SSA), which employs the structural properties of the point spread function. This method maps the gray-scale distribution of the star image spot into a matrix. This matrix is further segmented into contiguous sub-matrices, referred to as sieves. Sieves comprise a finite number of pixels. These sieves are evaluated and ranked based on their degree of symmetry and magnitude. Every pixel in the image spot carries the accumulated score of the sieves associated with it, and the centroid is its weighted average. The performance evaluation of this algorithm is carried out using star images of varied brightness, spread radius, noise level, and centroid location. In addition, test cases are designed around particular scenarios, like non-uniform point spread function, stuck-pixel noise, and optical double stars. The proposed algorithm is compared with various long-standing and state-of-the-art centroiding algorithms. The numerical simulation results validated the effectiveness of SSA, which is suitable for small satellites with limited computational resources. The proposed algorithm is found to have precision comparable with that of fitting algorithms. As for computational overhead, the algorithm requires only basic math and simple matrix operations, resulting in a visible decrease in execution time. These attributes make SSA a fair compromise between prevailing gray-scale and fitting algorithms concerning precision, robustness, and processing time.

## 1. Introduction

Gyro-less attitude and attitude rate estimation techniques, using measurements from stand-alone microlight attitude sensors, are critical for realizing a low-cost micro-satellite. Out of various operational sensors available, star sensors are the most accurate [1]. Star sensors observe star positions to estimate the attitude of a satellite. This is done by converting the star image spots formed on the detector of the star sensor to star line-of-sight (LOS) vectors. In order to derive the star LOS vectors precisely, the centroid of the image formed on the detector needs to be determined with good accuracy [2]. The process of finding the centroid of the star image in the best possible way, despite noise in the image sensors, is termed centroiding. Centroiding is aimed at extracting the key features from the image and, thereby, reducing the acquired image to a more compact and manageable data set for all subsequent processing [3].

Over the last few years, many methods have been developed for determining the centroid of a star image. These can be broadly classified into two streams, based on the statistical methodology employed to locate the star centroid. Gray-scale centroiding techniques are based on determining primary or secondary moments of the accumulated gray. These methods are essentially based on weighed averaging of the relative distance from the star centroid, where the gray, or squared gray, values serve as the weights in the computation [4]. This includes widely used techniques, like Center of mass (COM) [2], Center of mass with thresholding [5], Iterative weighted center-of-gravity (IWCOG) [6], and Squared gray-scale [7]. These algorithms are essentially averaging techniques that, eventually, provide a space where the star centroid can most likely be spotted. Hence, comparatively, they are less accurate and require less processing.

The alternate set of centroiding algorithms can be broadly described as curve fitting-based techniques, which rely on the energy distribution function of the star image spot incident on the image detector. The image of a bright point object in a dark background, obtained from a diffraction-limited optical system, consists of a bright central spot surrounded by a series of concentric rings [8,9]. This is termed “Airy disk” [10]. This matches the diffraction pattern obtained from the optical system of an ideal star sensor consisting of a well-focused, circular aperture lens assembly and the star image lying near its optic axis. Assuming aberrations and discrete sampling errors to be absent [11], this pattern can be approximated as a Gaussian function [12], especially if some slight, intentional defocusing is introduced, as shown in Figure 1.

The intensity distribution on the image detector plane resulting from a point source is termed the point spread function (PSF). It can have varied shapes depending on the associated optics. The equation of the PSF, which can be approximated as a Gaussian function, is given by,
(1)$$I\left(h,v\right)=\frac{I_{0}}{2\pi \sigma_{h}\sigma_{v}}\exp\left[\begin{array}{c} \frac{-{\left(h-h_{c}\right)}^{2}}{2\sigma_{h}^{2}} \end{array}\right]\exp\left[\begin{array}{c} \frac{-{\left(v-v_{c}\right)}^{2}}{2\sigma_{v}^{2}} \end{array}\right]$$
where I(h,v) is the gray intensity value at a pixel with sensor array coordinates (*h*,*v*); $\sigma_{h}$ and $\sigma_{v}$ are the spread of stellar intensities in horizontal and vertical directions, respectively; $\left(h_{c},v_{c}\right)$ are the array coordinates of the centroid point in horizontal and vertical directions, respectively; $I_{0}$ is the peak intensity of a particular star in the sensor array, measured in photoelectrons. The curve fitting-based centroiding techniques rely on developing the Gaussian curve that fits the best to the photo-electron distribution of the star image spot. This curve can be uniquely described with five Gaussian parameters, namely $I_{0}$, $h_{c}$, $v_{c}$, $\sigma_{h}$, and $\sigma_{v}$. Alternatively, this technique can be understood as an optimal mechanism to determine the best estimates of the Gaussian parameters. Well-established centroiding techniques, like Gaussian best fit (GBF) [13], and Gaussian three-point centroiding (G3P) [14], are curve fitting-based methods. Theoretically, this class of techniques is superior to gray-scale methods in estimating the star centroid with precision.

Various approximations and truncation errors, arising from discretization and sampling window limitations, result in systematic errors in centroiding [15]. An important component of systematic centroiding errors, namely the S-curve error [16], is found to limit the accuracy of gray-scale methods, but has a negligible effect on fitting methods. Nevertheless, fitting methods suffer from computational complexity as the solution of Gaussian parameters is fundamentally a nonlinear optimization problem. In addition, these techniques are iterative and are invariably sensitive to the initial conditions of Gaussian parameters.

Recently, a renewed interest has emerged in addressing these drawbacks. Several researchers have developed methodologies superior to conventional fitting-based centroiding techniques in precision and processing time. Gaussian grid algorithm [17] tries to estimate the best fit by converting it into a constrained minimization problem and deducing explicit equations for the solution of Gaussian parameters. This deduction suffers from the approximations from least square estimation and truncation of the Taylor series. Gaussian analytic centroiding (GAC) [4] takes an alternative path by developing a closed-form solution for Gaussian parameters by observing the marginal gray-scale accumulation in a horizontal and vertical direction. However, this algorithm is sensitive to temporally and spatially varying noise processes in the sensor array. This is attributed to the fact that this method solely depends on the photo-electron counts of the light-exposed pixels in the region of interest (ROI), irrespective of its signal-to-noise ratio. For real-time applications, calculating Gaussian parameters in closed form without compromising accuracy is achieved, to an extent, using the Fast Gaussian Fitting (FGF) [18] algorithm. It is an iterative feedback-based methodology that initially estimates the signal-to-noise ratio of comparatively brighter pixels to designate the ROI. Further, a closed-form linear equation set must be solved to obtain the Gaussian parameters. This algorithm suffers from comparatively longer processing time due to its iterative nature.

Designating the centroids of the star images formed on the sensor array to the highest possible accuracy becomes vital as it directly affects the attitude estimation accuracy. The maximum achievable attitude accuracy, imposed by theoretical limits, for our stellar neighborhood, reported in [19], suggests a possible improvement in the precision of current star sensors by about seven orders of magnitude. Optimizing the parameters to aid in the localization of the center of a star image plays a crucial role in moving closer to that limit. The star image formed on the sensor array is adulterated with various noise processes. These noise processes are classified as the following, based on their origins: equipment noise and background noise. Optimizing the optical assembly to limit the displacement of star images, particularly imprinted at the edge of the FOV frames in [20], and predicting systematic errors using machine learning in [21], are two of the recent developments to compensate for equipment fidelity. To account for temporarily bright pixels in the sensor array, ref. [22] used column offset corrected background-subtracted pixel values for centroiding. In high-agile missions, the image trail from the satellite’s motion can be shortened, to an extent, by optimizing the average exposure time [23]. However, it reduces the energy received at the detector. Ref. [24] introduces a difference hash algorithm, followed by block threshold segmentation, to extract the photo-electrons corresponding to a star from its image moving in an angular rate of the order of 3${}^{{}^{\circ}}/s$.

The complex starry sky background poses an additional challenge, as extracting the photo-electrons corresponding to a specific image spot is difficult in this context. Various image processing-based algorithms have been developed to mitigate this issue. The background estimation algorithm [25] facilitates centroid extraction by removing the background of the star image, though with additional computational overhead. Several denoising techniques have been developed recently to pre-filter the star images to preserve their shapes. Methodologies to ward off the interference of the background and enhance the star image-background contrast include the following: block adaptive threshold segmentation (BATS) [26], ring filter [27], hybrid filter [28], dark channel filter [29], and Improved Gaussian Side Window Filter (IGSWF) [30]. In the case of star sensors with small field-of-view (FOV), even muddled star image spots must be included in the attitude estimation process. In such cases, the improvement in star-centroiding accuracy of blurred image spots, owing to the star tracker’s dynamic operating environment, is discussed in [31].

Recent high agile and microlight satellite missions [32,33,34] reveal the trend towards the miniaturization of star sensors. Advances in the semiconductor industry further propel this swing. With the advent of high-precision CMOS active pixel sensors (APSs) and high-speed microcomputer system-on-chip (SOC) technologies, centroid accuracy and the threshold for detection have been increased manifold [35,36]. Henceforth, the photo-electron counts from the sensor arrays are treated more reliably. This paves the path to pixel gray value processing routines that consider the radial and uniform nature of the PSF. We propose an algorithm that dissects the ROI into sub-squares and individually treats the gray accumulation in them. This is called the Sieve Search Algorithm (SSA), owing to the sieve-like sub-squares in the ROI, which forms the search space for finding the centroid of the star spot.

In an exactly focused image, a star appears as a point source. However, the spread of energy resulting from that star is not confined to a single pixel, owing to diffraction. Nevertheless, the photo-electrons generated in the pixels adjacent to the most illuminated pixel are not sufficient to facilitate the centroiding process. Hence, the optics in the front end of the star sensor are intentionally defocused, resulting in the star image occupying several pixels. This assists the calculation of the center of the star to sub-pixel accuracy.

In this paper, we propose an algorithm that aids sub-pixel localization of a star image spot using the radially symmetric characteristics of the PSF. This algorithm analyzes the accumulated gray value corresponding to the image spot and implements a fast-converging averaging function to designate the radial symmetry center. This methodology is compared with algorithms [2,4,6,13,14,18] based on its sensitivity to various operating conditions and attributes of PSF. The proposed algorithm is found to give accuracy and noise robustness comparable with that of curve-fitting algorithms. The execution time required for the proposed algorithm is about half that of the most efficient Gaussian fitting algorithm available in the literature.

This paper is organized as follows: in Section 2, we develop an algorithm to locate the centroid of an image spot using contiguous sieves inside the ROI. In Section 3, we give a brief overview of methods and materials used in the experiments to evaluate the performance of the proposed algorithm. In Section 4, the results and analysis of the comparison of the performance of the proposed algorithm with various long-established and state-of-the-art centroiding techniques are discussed.

## 2. Sieve Search Algorithm

### 2.1. Characteristics of PSF Gray Value Distribution

Ideally, the luminous intensity distribution of a star image spot is radially symmetric about its centroid. It has been noted that the PSF’s symmetric nature deteriorates with the distance of the star spot from the optical axis and the asymmetric aberrations present in the optical assembly of the star sensor. Various techniques have evolved to treat these deformations in the PSF by measuring the star profile [37]. Hence, without loss of generality, we assume that the gray accumulation at the surface of a sensor array, resulting from a star spot in the night sky, is approximated to be a symmetric two-dimensional discretized Gaussian function. This is depicted in Figure 2a.

Let this Gaussian contour span across an n×n window on the sensor array. This is referred to as the region of interest (ROI). Figure 2b shows the 9×9 ROI corresponding to the PSF in Figure 2a. By mapping the accumulated gray value at each pixel in this ROI, a matrix is formed. This is referred to as the ROI matrix. Figure 3 presents the 9×9 ROI matrix formed from the PSF in Figure 2a.

The principal diagonals of the ROI matrix are the leading diagonal (spanning from the top left corner of the ROI matrix to the bottom right), denoted by $d_{L}$, and the counter-diagonal (from the top right corner to the bottom left), denoted by $d_{C}$. On observing the elements of this ROI matrix, it can be noted that:the ROI matrix is nearly symmetric. For an n×n ROI matrix, let $G_{L}$ represent the *n*-element vector, consisting of gray values in the pixels along the leading diagonal $d_{L}$. The off-diagonals $d_{LU}$ and $d_{LD}$ are *a* pixels apart from $d_{L}$. The gray value vectors corresponding to $d_{LU}$ and $d_{LD}$ are $G_{LU}$ and $G_{LD}$, respectively. It can be observed that the value of the $b^{th}$ elements in $G_{LU}$ and $G_{LD}$ are almost similar, where bϵ[1,n−a]. In Figure 3, the off-diagonals $d_{LU}$ and $d_{LD}$ are two pixels apart from $d_{L}$. The grey value vectors $G_{L}=\left[2\;6\;29\;94\;149\;115\;43\;9\;3\right]$, $G_{LU}=\left[4\;19\;63\;104\;83\;33\;7\right]$, and $G_{LD}=\left[5\;22\;68\;104\;77\;28\;6\right]$. The values of $b^{th}$ elements in $G_{LU}$ and $G_{LD}$ are found to be comparable, where bϵ[1,7]. In other words, the pixels along the off-diagonals, equidistant from a diagonal element on both sides of the principal diagonals, have comparable gray value accumulation.as we transverse along the pixels at the same radial distance from the brightest pixel, the magnitude increases as we move closer to the principal diagonal pixels. Here, magnitude is referred to the gray value of the pixel. Let $g_{c}$ represent the gray value of a pixel $p_{c}$, along $d_{L}$, at some radial distance from the most illuminated pixel in the ROI. At the same radial distance, in the neighborhood of $p_{c}$, the gray value of the pixels along $d_{LU}$ and $d_{LD}$ are $g_{bu}$ and $g_{bd}$, respectively. Then, $g_{c}$ is marginally greater than $g_{bu}$ and $g_{bd}$. In Figure 3, the brightest pixel has an accumulated gray value of 149. Among the pixels at the same radial distance from it, the gray value at the pixel along the principal diagonal {43} is marginally greater than that along the off-diagonal pixels {28,33}.

These observations are valid for both the principal diagonals, fostering our assumption that the PSF of the star image is radially symmetric. The extent of the veracity of these observations is verified in the subsequent subsections.

### 2.2. Sieve Segmentation and Symmetry

The ROI matrix is segregated into a finite number of reduced-order sub-matrices for ease of processing. These sub-matrices are referred to as sieves. A square sub-matrix of size *m* is described, starting from pixel (1,1) at the top left. This is designated by sieve $S_{1}$. The next m×m sub-matrix $S_{2}$ is described horizontally from pixel (1,2) to the right. Subsequently, sieves up to $S_{n-m+1}$ are generated from the first row. On completion of the first row, the next sieve $S_{n-m+2}$ is constructed from pixel (2,1), and the process continues. This way, $k^{2}$ sieves can be generated from an n×n ROI matrix, such that k=(n−m+1).

Among these segmented sieves, the ones having diagonal entries that coincide with any of the two principal diagonals of the ROI matrix are bound to have maximal symmetry and, hence, possess the highest chance to contain the centroid. These sieves are termed potential sieves. Consequently, the degree of symmetry of a sieve can be considered a metric to verify whether it is a potential sieve. In order to examine the extent to which a sieve Sq is symmetric along its leading diagonal, two matrices, namely, $S_{q_{sym}}$ and $S_{q_{asym}}$, are derived, such that,
(2)$$S_{q_{sym}}=S_{q}+S_{q}^{T}$$
and,
(3)$$S_{q_{asym}}=S_{q}-S_{q}^{T}$$

The degree of symmetry $\phi_{q}$ of sieve $S_{q}$ is determined by the following equation,
(4)$$\phi_{q}=\frac{norm\left(S_{q_{sym}}\right)-norm\left(S_{q_{asym}}\right)}{norm\left(S_{q_{sym}}\right)+norm\left(S_{q_{asym}}\right)}$$

In order to verify the degree of symmetry of sieve $S_{q}$ along its counter diagonal, it is rotated 90${}^{{}^{\circ}}$ clockwise. This rotated sieve is denoted by $S_{q[+90^{\circ }]}$. Equations (Equation 2)–(Equation 4) are applied on $S_{q[+90^{\circ }]}$ to determine its degree of symmetry $\phi_{q[+90^{\circ }]}$.

While transversing along the $p^{th}$ row of the ROI matrix, the index numbers of the potential sieves *u*, with respect to the principal leading diagonal, are given by,
(5)u=k×(p−1)+p
Similarly, the potential sieve index number *v*, corresponding to the principal counter diagonal, is given by,
(6)v=k×(p−1)+n−m−p+2

For instance, consider the ROI matrix in Figure 3. Let us assume the sieve size to be m=3, which results in 49 sieves. The potential sieves are designated by index numbers u=[1,9,17,25,33,41,49] and v=[7,13,19,25,31,37,43], respectively. These sieves have the highest degree of symmetry compared to the rest of the sieves in the ROI. This is illustrated in Figure 4a,b, where the degree of symmetry ϕ and $\phi_{\left[+90^{\circ }\right]}$ of all the sieves $S_{i}$ and $S_{i[+90^{\circ }]}$ in the ROI are plotted, respectively, where $i\;\epsilon \;1,\ldots ,k^{2}$.

### 2.3. Characteristics of Sieve Magnitude

Centroiding is defined as the sub-pixel localization of the intensity of a star image incident on the sensor array. The image spot’s centroid lies close to pixels with peak gray values. Consequently, sieves with comparatively higher magnitudes are more likely to contain the centroid. There are three primary ways to measure a matrix’s magnitude (μ): maximum coefficient, matrix norm, and maximum expansion [38]. The norm has a nearly uniform contribution from each element in the matrix, making it the most suitable measure to quantify and denote the magnitude of a matrix. The norm of the sieves in the ROI are plotted in Figure 5. On observing the plot of the trace of the sieves and the rotated sieves, they are found to approximately follow the norm curve and to coincide with it at data points corresponding to potential sieve index numbers *u* and *v*. Hence, in order to lessen the complexity involved in the computation of the matrix norm, the trace of the sieves and rotated sieves are used to signify the sieve magnitude μ and $\mu_{\left[+90^{\circ }\right]}$, respectively. The steps involved in centroiding, using the sieve search algorithm, are discussed in the following subsection.

### 2.4. Centroiding Based on Sieve Search Algorithm

As the star sensor is exposed to the night sky, star images form on the imaging array. Due to the intentional defocusing of the lens assembly, the star image spot occupies a larger pixel window. From the resulting PSF, the photo-electron counts, corresponding to each pixel in the star image, are extracted. The proposed star-centroiding algorithm is outlined as follows:(i)An ROI of size not less than 5×5 pixel window is created around the star image spot. The accumulated gray value distribution inside the ROI is used to populate the elements of the corresponding ROI matrix.(ii)The ROI matrix is segregated into square sub-matrices called sieves. The degree of symmetry and magnitude of each sieve $S_{i}$ in the ROI is evaluated and stored as $\phi_{i}$ and $\mu_{i}$, respectively, where $i\;\epsilon \;1,\ldots ,k^{2}$. The sieves are rotated by $90^{\circ }$ clockwise. The corresponding $\phi_{i\left[+90^{\circ }\right]}$ and $\mu_{i\left[+90^{\circ }\right]}$ are determined.(iii)The sieve probability index $P(S_{i})$, evaluated for each sieve $S_{i}$ in the ROI according to Equation (Equation 7), gives the probability of a sieve to containing the centroid location of the image spot. This is depicted in Figure 6.
(7)$$\rho_{i}=\left(\phi_{i}+\phi_{i\left[+90^{\circ }\right]}\right)\cdot \left(\mu_{i}+\mu_{i\left[+90^{\circ }\right]}\right),\;\;\;\;\;\;\;\;\;P\left(S_{i}\right)=\frac{\rho_{i}}{max(\rho_{i})}$$(iv)For every sieve $S_{i}$ in the ROI, the value of its degree of symmetry–magnitude product $\rho_{i}$ is associated with each pixel contained in it. Consequently, for a pixel *j*, the value of $\rho_{i}$, resulting from the various sieves it constitutes, adds up, given by,
(8)$$\psi_{j}=\sum_{i\epsilon U}\rho_{i}$$
where *U* is a set formed by the sieves that contain pixel *j*. As is depicted in Figure 7, the probability of each pixel in the ROI possessing the centroid of the image spot, referred to as the pixel probability index P(j), is given by,
(9)$$P\left(j\right)=\frac{\psi_{j}}{max(\psi_{j})}$$(v)Let $h_{j}$ and $v_{j}$ correspond to the horizontal and vertical coordinates of pixel *j*. The value of the centroid is computed as,
(10)$$h_{c}=\frac{\sum_{j\epsilon W}h_{j}\cdot \psi_{j}}{\sum_{j\epsilon W}\psi_{j}}\;\;\;\;\;v_{c}=\frac{\sum_{j\epsilon W}v_{j}\cdot \psi_{j}}{\sum_{j\epsilon W}\psi_{j}}$$
where *W* is a set formed by the pixels in the star spot. A flowchart explaining the working of the SSA is given in Figure 8.

### 2.5. Optimal Sieve Size

The influence of the size of the sieve on the centroiding accuracy and algorithm execution time is discussed in this section. As seen in Figure 9a, the transient centroiding error $e_{c}\left(0\right)$ increases drastically with increase in the size of the sieve. This is attributed to the insufficiency of terms in the averaging function given in Equation (Equation 10), as the size of the sieve increases. For all the sieve sizes, the steady-state value of the centroiding error, to which the algorithm converges $e_{c}\left(\infty \right)$, is approximately the same. The time complexity involved in the centroid computation using SSA is of the order of $O{\left(n-m+1\right)}^{2}$. Hence, as seen in Figure 9b, the computation time decreases with an increase in the size of the sieve. For a 9×9 ROI, the sieve size m=3 is the optimal choice, as, at this value, the algorithm gives the minimal transient error and converges rapidly.

## 3. Materials and Methods

The performance of the proposed algorithm was evaluated by testing it on a set of digitally-simulated star images, having varied characteristics, and comparing the results with existing centroiding techniques. The generation of synthetic star images, selection of attributes, and the details of the tests conducted to evaluate the performance are discussed in the following sub-sections.

### 3.1. Simulation of Star Images

The attitude matrix of a nadir-pointing satellite is modeled with the star sensor camera directed along its roll axis. The STAR 1000 APS CMOS sensor array, originally manufactured by FillFactory NV, Belgium [39], was selected as the image sensor. The focal length of the optical assembly was 50 mm, the f-number of the lens was 1.3, and the FOV of the camera was 17${}^{{}^{\circ}}$. It was assumed that the PSF, formed by defocusing the lens, was a 2-D Gaussian function with an equal spread in two directions.

With the help of a modeled attitude matrix, the stars visible on the imaging array at a particular time instant were simulated by computing the angle subtended by each star with the optical axis (boresight) at the focus. The number of photo-electrons reaching the sensor array for a star of visual magnitude 0 within a wavelength of 400–800 nm, with a lens aperture of 1 mm${}^{2}$, was 19,100 [2]. So the number of photo-electrons that reached the imaging array for a star of visual magnitude 0.5, with 30 ms integration time and having lens diameter (f/f-number= 50 mm/1.3) 38.46 mm, is,
(11)$$\begin{array}{ccc} L_{0} & = & 19100\left(\frac{1}{2.5^{0.5-0}}\right)0.03\pi {\left(\frac{38.46}{2}\right)}^{2}\\ \; & = & 4.2\times 10^{5}photoelectrons/exposure. \end{array}$$

Only a part of these photons is converted into photo-electrons by the array. This depends on the quantum efficiency and fill-factor of the image sensor. The quantum efficiency-fill factor product, according to the STAR 1000 datasheet [39], was 30%.

### 3.2. Simulation of Noise Process

The error due to quantization and the background noise floor was included in the simulation of the sensor array to emulate reality. The background noise floor includes dark current noise, photon shot noise, photo response non-uniformity, fixed pattern noise, and read-out noise. The various noise processes were modeled according to the STAR 1000 data sheet [39]. This amounted to around 7% of the photons accumulated in the most illuminated pixel in the star image spot. The energy distribution, due to star spot intensities on the STAR 1000 APS CMOS sensor array at a particular instant of time, is depicted in Figure 10.

### 3.3. Selection of Attributes

Unless specified otherwise, the following were the attributes of the PSF used in the experiments for testing the algorithms in the subsequent section.

the radius of the Gaussian spread in both directions, $\sigma_{PSF}=0.75$.the size of the star image pixel window was 3×3.the average level of noise corresponding to a star image spot was restricted to 7% of the accumulated gray value in the most illuminated pixel of that star image.

### 3.4. Particulars of Experiments

The sensor assembly simulated in the previous subsections was mounted on a nadir-pointing satellite moving at an angular velocity of 0.0011 radians-per-second in all three directions. As the satellite moves, the star image spots of varied radiance form at different locations on the sensor array. The data update rate was taken as 10 Hz. The algorithms for centroid estimation were tested on the stars in the output image for 10,000 s, and its mean was considered the centroiding error. Simulations were carried out in a 12th Gen Intel(R) Core(TM) i7-12700H CPU, 16 GB RAM, and 64-bit operating system with MATLAB version 2022b. The performance evaluation results of the proposed algorithm are explained in the following section.

## 4. Results and Analysis

Various centroiding algorithms [2,4,6,13,14,18], that are currently available in the literature, and those employed in flight-worthy sensors, were compared with SSA for varied scenarios, and this is explained in this section.

The COM algorithm was applied to the pixels after subtracting average background noise over the pixels in a 5×5 pixel window. For IWCOG, the weighing Gaussian spread was assumed to be distributed over 1 pixel in both directions. For FGF, the signal-to-noise ratio was calculated for a pixel window of 3×3.

### 4.1. Sensitivity of Centroiding Accuracy to Various Parameters

By centroiding accuracy, we refer to the Euclidean separation distance between the calculated centroid and the actual centroid. The sensitivity of the centroiding accuracy, with respect to the variation of various attributes of the PSF, lens assembly, and image spot, is verified in the following subsections.

#### 4.1.1. Brightness

The variation in the accuracy of the centroiding algorithms with respect to the changes in the brightness of the image spot was studied in this experiment. This was done with stars of visual magnitude ranging from −1.46 to 5.5. The error curve is shown in Figure 11.

When a bright star is imprinted on the sensor array, numerous pixels in the PSF get saturated. The algorithms GBF, G3P, and FGF have an inherent design to reject the saturated pixels. However, for stars in the brightness range between visual magnitude −1.46 to −1.1, the gray information passed over from the PSF for centroiding after rejecting the saturated pixels is not enough to resolve the centroid. Hence, the centroiding error is high for these algorithms in this interval. As seen in Figure 11, in this regime, the truncation of photon count resulting from saturation also affected the accuracy of COM, IWCOG, GAC, and SSA. For stars fainter than −1.1 VM and brighter than 3.4 VM, the curves corresponding to GBF, G3P, FGF, GAC, and SSA were stable. Though less so initially, the accuracy of COM and IWCOG improved for stars between brightness −1.1 VM and −0.45 VM. Further, the accuracy of these two algorithms was found to follow a stable trend until 2.2 VM. For stars fainter than 2.2 VM, the centroiding error was found to be considerably high. This is because, once the star brightness drops beyond a particular value, the effect of photo-electrons from noise processes becomes a prominent factor in accurate measurements. In the cases of GBF, G3P, FGF, GAC, and SSA, this point was found to be 3.4 VM. The centroiding error was high for all the algorithms in this operating regime. The sensitivity and magnitude of the centroiding error from the proposed algorithm were comparable to the fitting algorithms for the complete range of brightness.

#### 4.1.2. Location of Star Spot Inside a Pixel

The sensitivity of the centroiding accuracy to the location of the centroid inside a pixel was evaluated in this experiment. For this, the location of the centroid of the star image spot was moved along the *x*-axis from one end of the pixel to the other. It is noted that the relation between star centroid location and centroiding accuracy varies with the radius of the Gaussian spread of the star spot. Hence, the Gaussian radius was varied uniformly between 0.5 and 1. The error curves are shown in Figure 12.

As seen in Figure 12d, the sensitivity of centroiding accuracy to the location of the star centroid inside a pixel was found to be negligible for algorithms GBF, G3P, and FGF. This variation was unaffected by the changes in Gaussian radius $\sigma_{PSF}$ for these algorithms. The variation of centroiding accuracy with the location was periodic for GAC (Figure 12a). For the algorithms COM (Figure 12b), IWCOG (Figure 12c), and SSA (Figure 12e), the variation was in the form of a sine-curve. It is seen that there are three locations where the centroiding accuracy was found to be at a minimum: at the center and at each end of the pixel. This is attributed to S-curve errors exclusively found in gray-scale algorithms. It is to be noted that for these three algorithms, the majority of the data points are centered around the minimum centroiding error. This corresponds to the centroiding accuracy for Gaussian radius $\sigma_{PSF}>0.75$. As seen in (Figure 12e), this variation in centroiding accuracy, due to S-curve errors, was minimal for SSA, compared to COM and IWCOG.

#### 4.1.3. Noise Floor

The changes in the centroiding accuracy according to the variation in the level of noisy photo-electrons in a star image were analyzed in this experiment. The noise floor was varied from 7% to 44% of the gray value accumulated at the most illuminated pixel in the star image spot. The error curves are shown in Figure 13.

As seen in Figure 13, the centroiding error increased with an increase in noise level for all the algorithms. The vulnerability to noise variations was highest for IWCOG, followed by GAC and COM. GBF displayed the slightest centroiding error to noise level variations. The noise robustness of the proposed algorithm was found to be equivalent to those of FGF and G3P.

#### 4.1.4. Gaussian Spread

The sensitivity of the centroiding accuracy to the changes in the Gaussian radius of the image spot was studied in this experiment. The Gaussian radius $\sigma_{PSF}$ was varied uniformly between 0.75 to 1.5. As seen in the error curves shown in Figure 14, the algorithms COM, IWCOG, and SSA were found to have centroiding errors resulting from S-curve errors until $\sigma_{PSF}\approx 0.9$. Beyond this point, the S-curve errors were found to reduce. The proposed algorithm’s error curve coincided with those of GBF, G3P, and FGF as the Gaussian radius increased. On the contrary, for COM and IWCOG, the error increased considerably as the value of $\sigma_{PSF}$ increased. This stems from the fact that the impact of the noise process is severely felt at pixels far away from the centroid, which is included in averaging as the spread radius increases. The error in GAC was high, owing to the periodic nature of centroiding error variation for $\sigma_{PSF}>0.75$, as depicted in Figure 12a.

### 4.2. Special Cases: Performance Evaluation

In this section, some special cases likely to be encountered by the star sensor camera during its operational lifespan are discussed. Ideally, the star PSF is assumed to be a symmetric two-dimensional discretized Gaussian function. However, the presence of aberrations and discrete sampling results in distortion of the PSF. As a result, the PSF is no longer symmetric, limiting it to spread non-uniformly with respect to its axes. This is depicted in Figure 15a.

During long exposure of the pixels in dim-lighting conditions, as in the night sky, some pixels in the ROI of the star image spot become saturated and are stuck at full well capacity [40]. This is called stuck-pixel noise and can produce undesirable artifacts in the centroid estimation. This is depicted in Figure 15b.

Optical double stars appear close together through chance alignment with the star sensor camera. The presence of optical doubles is not uncommon in the trajectory of a satellite. A star-pair which are apart by less than 10 arc-seconds, for instance HJ 1846 [41], was considered for this experiment. Figure 15c presents the pixel intensity of the image array corresponding to optical double stars that were 5 pixels apart from reach other.

The scenarios mentioned above were used as test cases, to further evaluate the performance of the proposed algorithm, and are described in the following subsections.

#### 4.2.1. Non-Uniform PSF

To verify the sensitivity of centroiding algorithms to non-uniform PSF, the Gaussian radius along the *y*-axis $\sigma_{PSF-Y}$ was increased uniformly from 0.5 to 1.5 pixels, and the spread along the *x*-axis $\sigma_{PSF-X}$ was fixed at 1.5 pixels. The error curves are shown in Figure 16. The centroiding accuracy of IWCOG, COM, GAC, and SSA were high until $\sigma_{PSF}\approx 0.9$. Beyond this point, the star image occupied more pixels, facilitating the centroiding process. The SSA algorithm achieved centroiding accuracy comparable to those of algorithms GBF, G3P, and FGF. RTandom noise was the major contributor to the centroiding accuracy in this interval, which severely impacted the accuracy of COM, IWCOG, and GAC.

#### 4.2.2. Stuck Pixel Noise

For imaging arrays with high ISO numbers, random pixels get stuck at full well capacity in due course of operation. The position of this hot pixel was varied in an ROI of size 15×15 to study the sensitivity of the centroiding algorithms. The center of the image spot was fixed at pixel location (8,8). The error curves are shown in Figure 17. As seen in Figure 17, the effect of a random saturated pixel on centroiding accuracy became significant as it transversed towards the star centroid. The COM and IWCOG were sensitive to a hot pixel anywhere inside the pixel window around the image centroid. This was due to the interference of an unusually high-intensity valued pixel in the averaging calculations. The algorithms GAC and FGF were similarly affected, though the magnitude of error was much less. Hot pixels only influenced the GBF and G3P once at the centroid pixel (8,8). This could be attributed to its inherent mechanism to discard a saturated pixel. The SSA was found to be vulnerable to stuck pixel noise when a hot pixel appeared anywhere inside the 3×3 pixel window around the image centroid. However, the resulting error was insignificant compared to those of the other algorithms, except for GBF.

#### 4.2.3. Optical-Double Stars

This study intended to evaluate the accuracy performance of the centroiding algorithms when subjected to a binary star pair less than one full-width-at-half-maximum apart. An optical-double star pair was modeled, with stars of different visual magnitudes and centroids separated by 5 pixels. The brighter star, the primary, had a 0.5 VM. The brightness of the fainter star, the secondary, was uniformly varied from 5.5 VM to 0.7 VM. This variation resulted in a change in the location of the brightness barycenter of the star pair given by,
(12)$$r_{c}=\frac{B_{1}\cdot r_{1}+B_{2}\cdot r_{2}}{B_{1}+B_{2}}$$
where $r_{1}$ and $r_{2}$ correspond to the centroids, and $B_{1}$ and $B_{2}$ correspond to the brightness intensities of the brighter and dimmer star in the pair, respectively. The brightness intensity of a star *i* was computed from its magnitude $m_{i}$ in the star tracker’s pass-bands, according to the equation given by,
(13)$$B_{i}=B_{ref}\cdot 10^{-0.4\cdot m_{i}}$$
where $B_{ref}$ is the light flux corresponding to a star of magnitude zero in the magnitude scale used by the star tracker. If the star tracker’s spectral response is very close to that of the human eye, then $m_{i}$ can be approximated to the visual magnitude of star *i*.

As the stars were close, there was no well-defined “valley” between the PSFs, making it challenging to resolve the centroid of an individual star in the pair without inference from the adjacent companion star [42]. Hence, we resorted to the estimation of the brightness barycenter of the pair. The variation of centroiding accuracy to the changes in location of the brightness barycenter was studied, and the error curve is depicted in Figure 18.

As seen in Figure 18, the brightness barycenter resided near the primary centroid when the secondary star was fainter than 4 VM. The algorithms COM, GBF, G3P, FGF, and the proposed algorithm resolved the centroid of the brightness barycenter with minimum error. The IWCOG and GAC had a visible centroiding error, due to their inability to resolve the sensor noise from the spilled-over secondary star’s photo-electrons. As the secondary’s brightness increased, the brightness barycenter of the pair shifted towards the secondary. In this scenario, only the proposed algorithm followed the actual centroid. The resulting centroiding error was minimal and was not more than 0.2 pixels in the complete operational regime.

### 4.3. Execution Time

The real-time computer-simulated execution time for the centroiding algorithms was evaluated and presented in terms of % in Figure 19. The algorithms COM, IWCOG, and GAC were the fastest. The GBF was found to be the most time-consuming algorithm. The G3P reduced the time required for Gaussian fitting by 21% when compared to GBF. The FGF was found to reduce the operational time further by 20%. The time required for the execution was less than half of the time required for FGF.

## 5. Conclusions

This study proposed a novel centroiding technique. termed the Sieve Search Algorithm (SSA), which uses the structural attributes of the PSF. The simulation of the sensor array STAR 1000 was used as a testbed for comparing the performance of the proposed algorithm with various long-standing and state-of-the-art centroiding algorithms. Star images with varied brightness, spread radius, noise level, and centroid location were used for the testing. Verification processes were carried out for special scenarios like non-uniform PSF, stuck-pixel noise, and optical double stars.

It is inferred that COM, though simple, is an inferior technique, as it includes noisy pixel intensities in the centroid calculation, and the background noise is assumed to be constant. A constant noise level does not reflect the spatially and temporally varying noise floor. In IWCOG, the Gaussian weighting function would not be able to change its centroid location and, hence, is susceptible to variation in Gaussian spread radius. The GBF is found to be computationally intensive as it demands closed-form solutions of nonlinear equations. The G3P gives a reasonable estimate, but demands the calculation of multiple logarithmic functions. The GAC, though computationally less intensive, is susceptible to noise-corrupted intensities. Hence, a separate feedback mechanism that considers the signal-to-noise ratio of the pixels in the ROI could result in a more effective estimation. The accuracy obtained from FGF is comparable with other fitting-based algorithms. However, the iterative feedback-based methodology suffers from computational complexity, as it relies on the solution of five linear equations and a complex feedback structure. The SSA was found to be susceptible to S-curve errors when the Gaussian radius of the image spot is less than 0.75.

In summary, the gray-scale centroiding methods (COM, IWCOG) and GAC are the fastest but only moderately precise and less robust to parameter variations and noise levels. On the other hand, the fitting algorithms (GBF, G3P, FGF) have better precision and stability in the vicinity of parameter variations, but require more time to execute. In most operational scenarios, SSA’s precision is comparable to fitting algorithms. The SSA requires only basic math and simple matrix operations, saving execution time. Hence, SSA can be seen as a fair compromise between prevailing gray-scale and fitting algorithms concerning precision, robustness, and processing time.

## Figures and Tables

**Figure 1 sensors-23-03222-f001:**
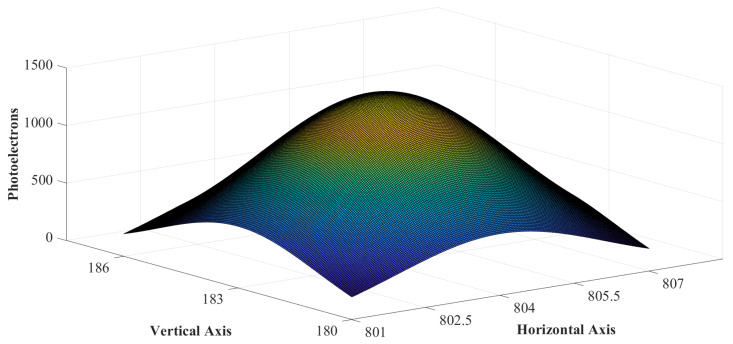
Energy distribution contour of a defocused star spot on the sensor array.

**Figure 2 sensors-23-03222-f002:**
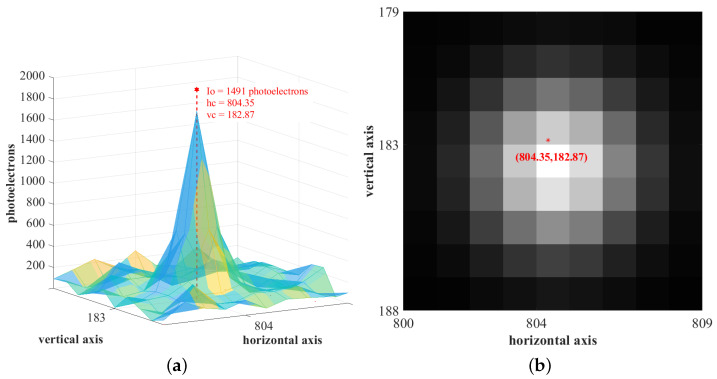
Pixel intensity of star image spot on a sensor array. The actual centroid of this star image is designated at (804.35,182.87). The pixel closest to the centroid is the brightest and generates 1491 photons. (**a**) The point spread function is (**b**) The gray-scale accumulation inside the ROI.

**Figure 3 sensors-23-03222-f003:**
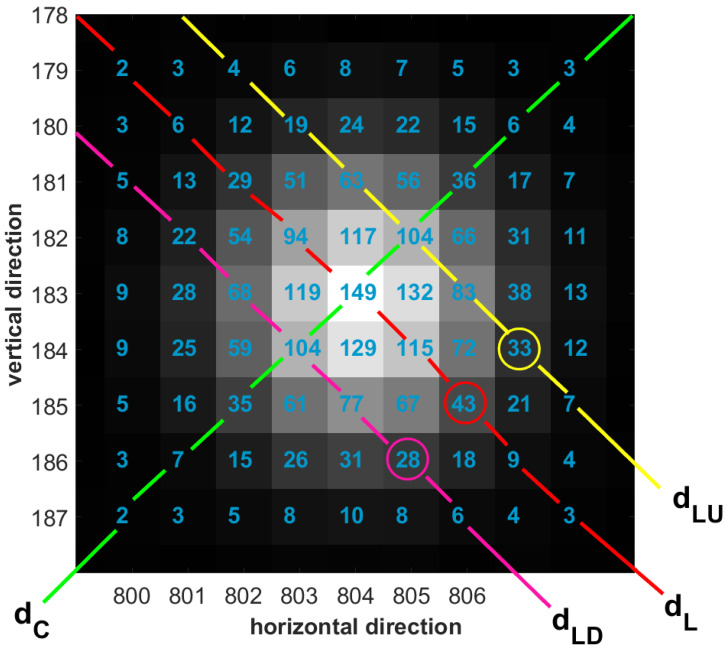
The 9×9 ROI matrix.

**Figure 4 sensors-23-03222-f004:**
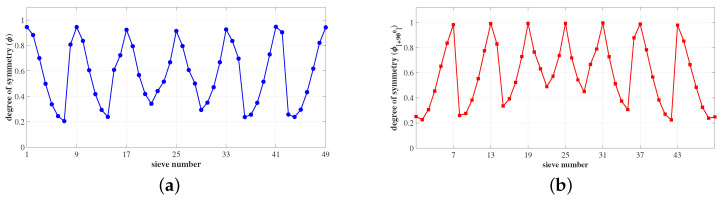
The degree of symmetry for all sieves $S_{i}$ and $S_{i[+90^{\circ }]}$ in the ROI. (**a**) ϕ of all sieves *S* in the ROI. ϕ peaks for sieves at index numbers u=[1,9,17,25,33,41,49]; (**b**) $\phi_{\left[+90^{\circ }\right]}$ of all sieves $S_{\left[+90^{\circ }\right]}$ in the ROI. $\phi_{\left[+90^{\circ }\right]}$ peaks at v=[7,13,19,25,31,37,43].

**Figure 5 sensors-23-03222-f005:**
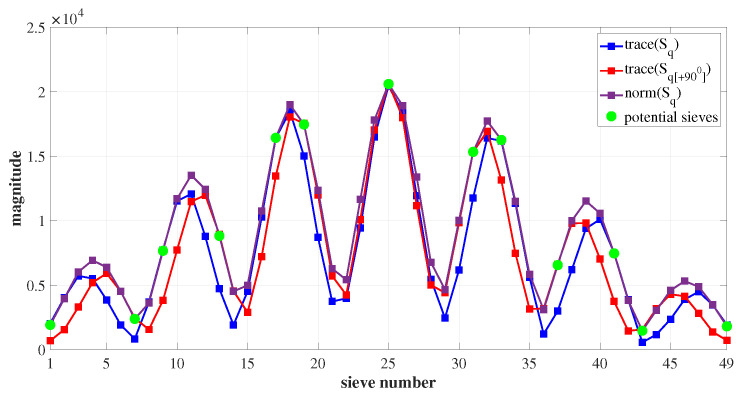
The norm; the trace of sieves and rotated sieves.

**Figure 6 sensors-23-03222-f006:**
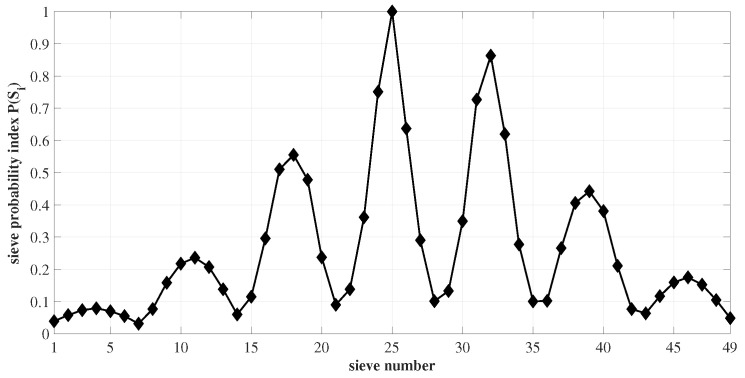
The sieve probability index $P(S_{i})$ of all the sieves in the ROI.

**Figure 7 sensors-23-03222-f007:**
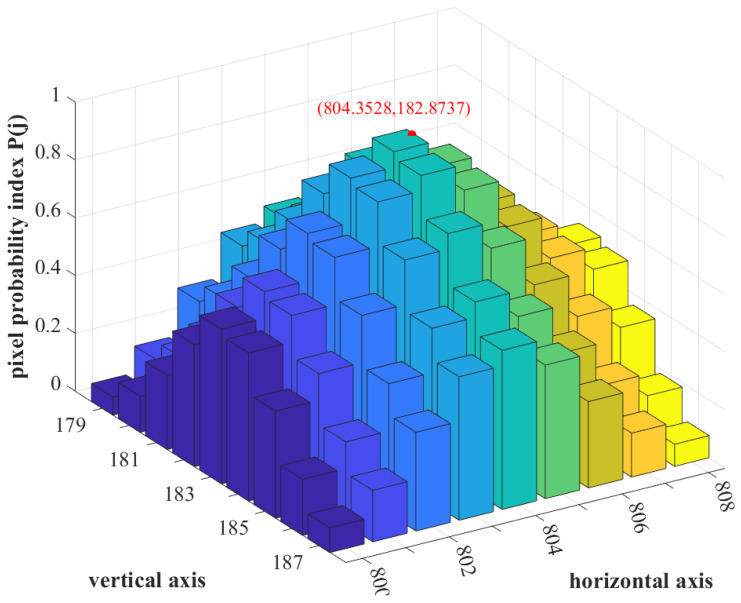
The pixel probability index P(j) of the pixels in the ROI.

**Figure 8 sensors-23-03222-f008:**
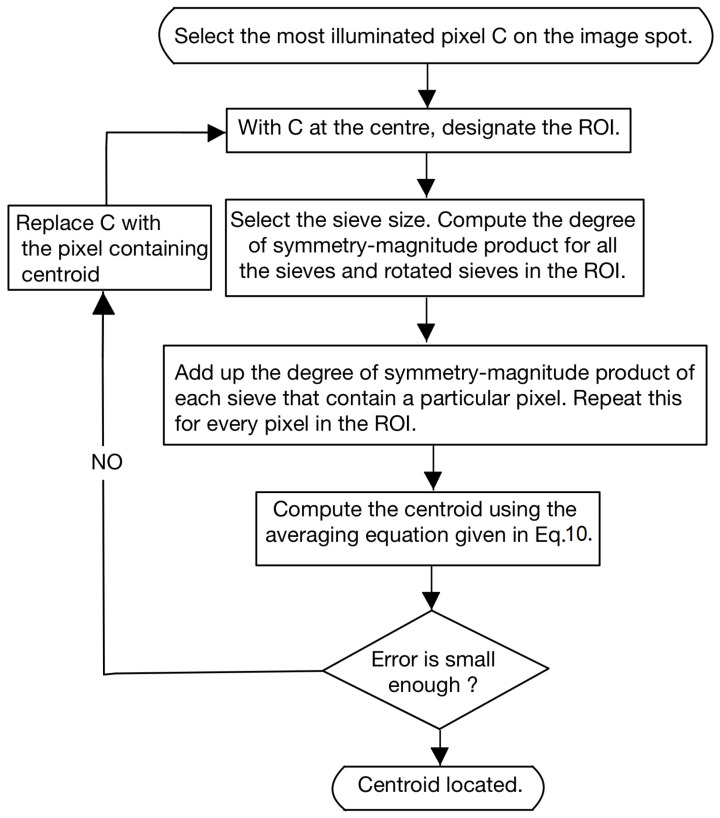
The flowchart for centroiding using SSA.

**Figure 9 sensors-23-03222-f009:**
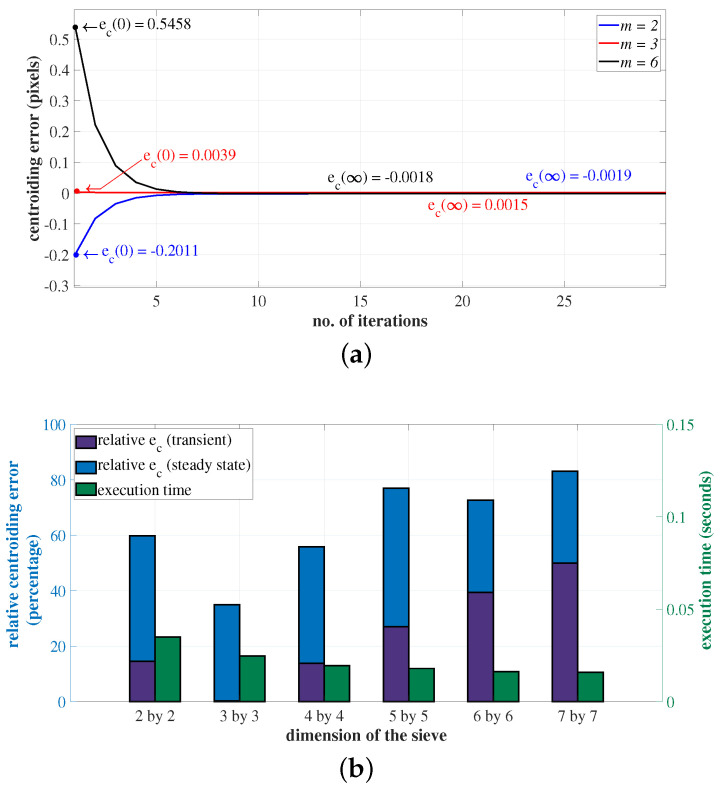
The optimal size of the sieve. (**a**) Transient and steady-state centroiding errors for sieve size m=2,3 and 6; (**b**) Comparison of various sieve sizes, based on the centroiding error and execution time.

**Figure 10 sensors-23-03222-f010:**
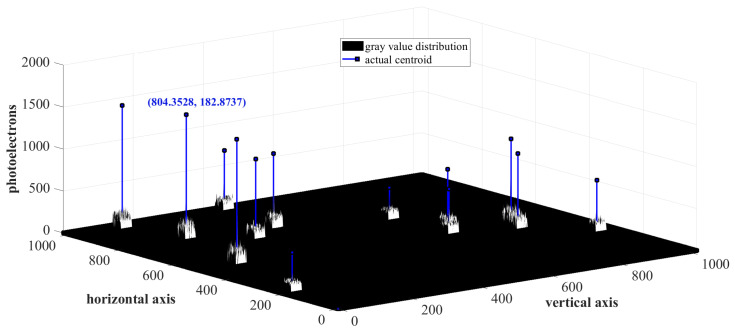
Pixel intensities of detected stars on STAR 1000 APS CMOS array.

**Figure 11 sensors-23-03222-f011:**
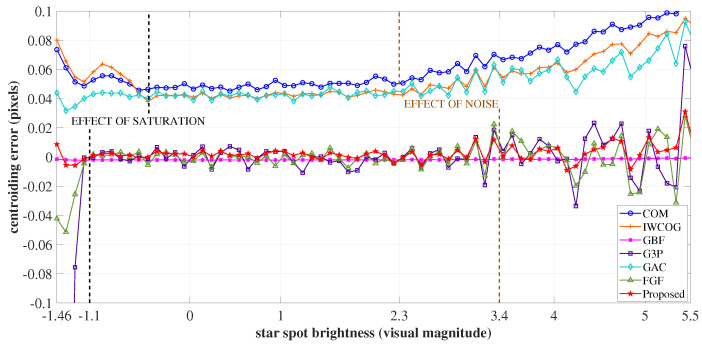
The variation of centroiding error with respect to the changes in the brightness level of the star spot. The SSA algorithm was comparatively stable throughout the operation regime.

**Figure 12 sensors-23-03222-f012:**
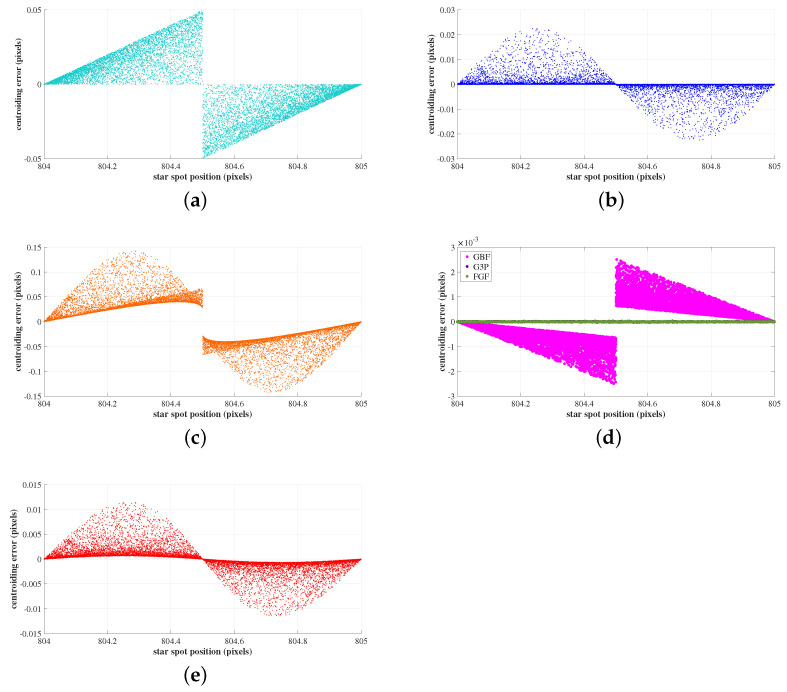
The variation of centroiding errors to changes in the location of the star image spot inside a pixel for various centroiding algorithms. The variation centroiding accuracy, due to S-curve errors, was minimal for SSA, compared to COM and IWCOG. (**a**) GAC; (**b**) COM; (**c**) IWCOG; (**d**) GBF, G3P and FGF; (**e**) SSA.

**Figure 13 sensors-23-03222-f013:**
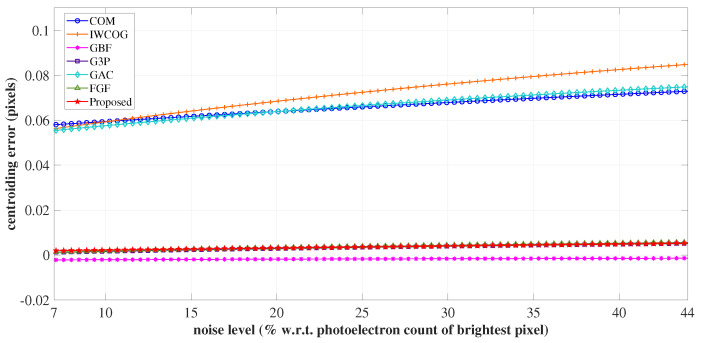
The variation of centroiding error with the changes in noise level. The noise robustness of SSA was found to be equivalent to that of FGF and G3P.

**Figure 14 sensors-23-03222-f014:**
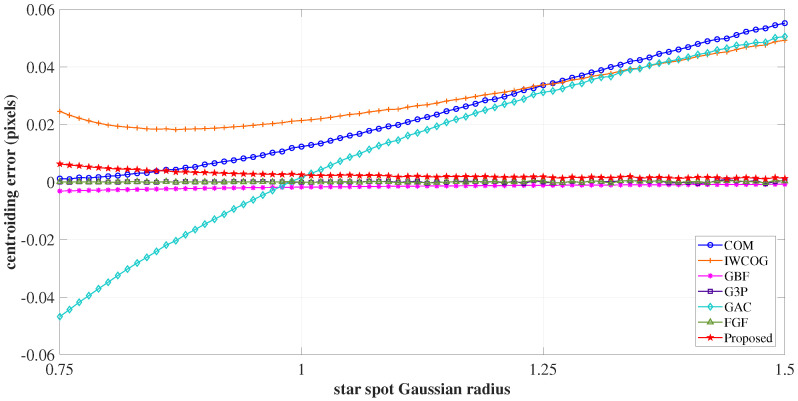
The variation of centroiding error with the changes in the Gaussian radius. The centroiding error in SSA was high until $\sigma_{PSF}\approx 0.9$. Beyond this, as the Gaussian radius increased, the centroiding accuracy of SSA coincided with those of GBF, G3P, and FGF.

**Figure 15 sensors-23-03222-f015:**
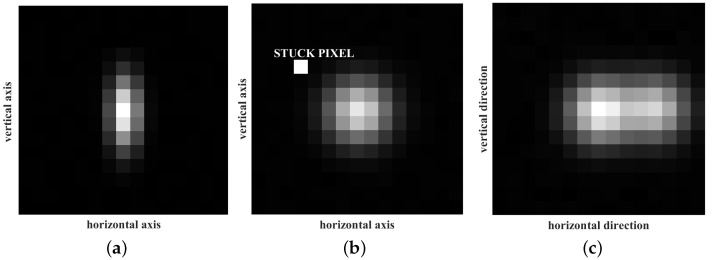
The special cases encountered by the star sensor camera during its operational lifespan. (**a**) Non-uniform PSF; (**b**) Stuck-pixel noise; (**c**) Optical double stars.

**Figure 16 sensors-23-03222-f016:**
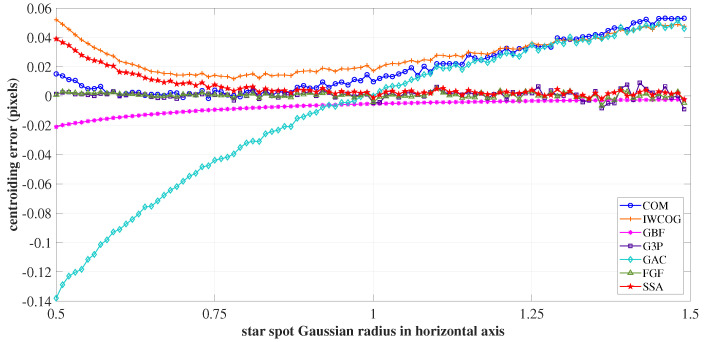
The variation of centroiding error with an increase in $\sigma_{PSF-Y}$ from 0.5 to 1.5. For $\sigma_{PSF-Y}>0.9$, the centroiding accuracy of SSA was comparable with those of GBF, G3P, and FGF.

**Figure 17 sensors-23-03222-f017:**
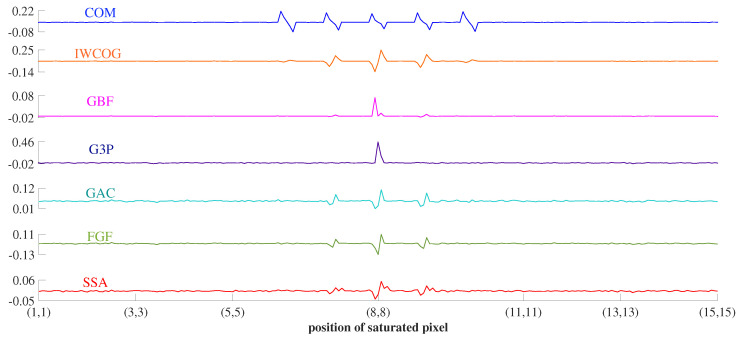
Sensitivity of centroiding accuracy to the location of stuck-pixel noise in the ROI. The magnitude of error in SSA was insignificant compared to the other algorithms, except for GBF.

**Figure 18 sensors-23-03222-f018:**
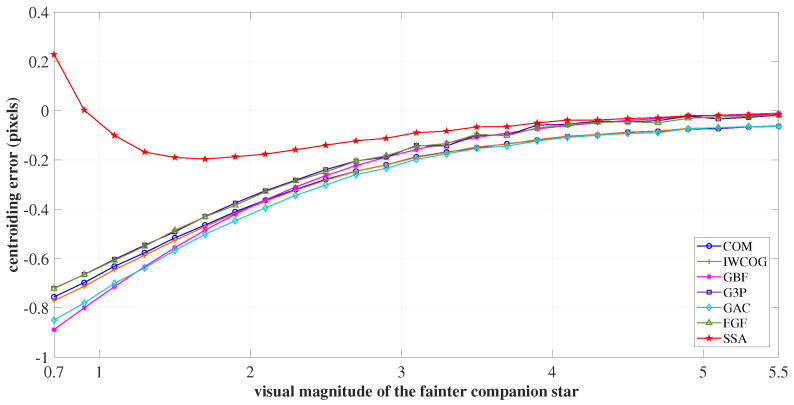
The centroiding error in resolving the visual magnitude barycenter of an optical double star pair. The SSA follows the actual centroid closely, resulting in minimum centroiding error.

**Figure 19 sensors-23-03222-f019:**
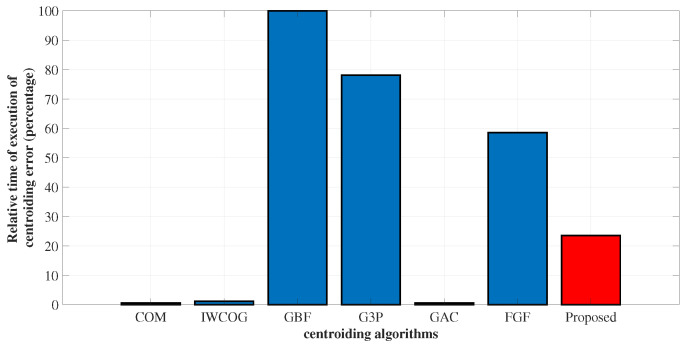
Comparison of relative real-time computer-simulated execution time for various centroiding algorithms. The gray-scale algorithms COM, IWCOG, and GAC were found to have the lowest execution time. SSA was found to require considerably less execution time in comparison with the curve fitting algorithms.

## Data Availability

Not applicable.

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
