# Peer review of "Sieve Search Centroiding Algorithm for Star Sensors"

_sensors, 2023, doi:10.3390/s23063222_

Round 1
Reviewer 1 Report (Previous Reviewer 1)
Please see my comments in the attached file review_v02.pdf.

Author Response
Please see the attachment.

Reviewer 2 Report (New Reviewer)
The localization of the center of the star image formed on the sensor array directly affects the attitude estimation accuracy. This paper proposes an intuitive self-evolving centroiding algorithm termed the sieve search algorithm (SSA), which employs the structural properties of the point spread function. This method maps the gray-scale distribution of the star image spot into a matrix. This matrix is further segmented into contiguous sub-matrices referred to as sieves. Sieves comprise a 5 finite number of pixels. These sieves are evaluated and ranked based on their degree of symmetry 6 and magnitude. Every pixel in the image spot carries the accumulated score of the sieves associated 7 with it, and the centroid is its weighted average. The performance evaluation of this algorithm is 8 carried out using star images of varied brightness, spread radius, noise level, and centroid location. In 9 addition, test cases are designed around particular scenarios like non-uniform point spread function, 10 stuck-pixel noise, and optical double stars. The proposed algorithm is compared with various 11 long-standing and state-of-the-art centroiding algorithms. The numerical simulation results validate 12 the effectiveness of SSA, which is suitable for small satellites with limited computational resources. The proposed algorithm is found to have precision comparable with that of fitting algorithms. As for computational overhead, the algorithm requires only basic math and simple matrix operations, resulting in a visible decrease in execution time. These attributes make SSA a fair compromise between prevailing gray-scale and fitting algorithms concerning precision, robustness, and processing time.
Results seem fine but there are several issues that should be address before the publication.
1) Star sensors observe star positions to estimate the attitude of a satellite. This is done by converting the star image spots formed on the detector of the star sensor to star line-of-sight (LOS) vectors. In order to derive the star LOS vectors precisely, the centroid of the image formed on the detector needs to be determined with good accuracy- add a reference for this statement.
2) The process of finding the centroid of the star image in the best possible way, despite the noise in the image sensors, is termed centroiding. Centroiding is aimed to extract the key features from the image and thereby reduce the acquired image to a more compact and manageable data set for all subsequent processing- add a reference for this statement.
3) Over the last few years, many methods have been developed for determining the centroid of a star image. This can be broadly classified into two streams based on the statistical methodology it employs to locate the star centroid. Gray-scale centroiding techniques are based on determining primary or secondary moments of the accumulated gray and the relative distance from the star centroid. - add a justification of this statement by adding some references.
4) Add the working of figure 3 in the text.
5) How the equations 2-4 are working for the segmentation process? add some justification in the form of text.
6) How the magnitude is computed? What is the input of magnitude equation?
7) What are the darksides of this work? add under the conclusion section.
8) In Fig. 8, how you realize that the error is small enough?
Round 2
Reviewer 2 Report (New Reviewer)
The authors well revised this version. It can be accepted in its current form.
Round 3
Reviewer 2 Report (New Reviewer)
The authors well address my comments.
This manuscript is a resubmission of an earlier submission. The following is a list of the peer review reports and author responses from that submission.
Round 1
Reviewer 1 Report
Please see my comments in the attached file: review_v01.pdf.

Reviewer 2 Report
The research work of the accuracy t of estimation of the exact center of the star image is meaningful for the estimation of the attitude and attitude rate of a spacecraft. A minor revision should be presented as follows.
1. The main contribution of the manuscript should be analyzed and discussed in detail. The flowchart of the proposed sieve search algorithm should be given.
2. Some typos and numerical expressions should be carefully corrected, such as “This gives the bounds of the space……the the peak value……” in page 6 and “The norm of vector R,plotted for all n−2×n−2 matrices……” at line 120 in page 7,etc.